# Awareness of COVID-19 influences on the wellness of Thai health professional students: An ambulatory assessment during the early "new normal" informing policy

**Krueakaew Tiaprapong, Achiraporn Sirikul, Chamawee Krajangmek, Namfon Duangthongkul, Nichaya Pandam, Nitita Piya-amornphan** *

Department of Physical Therapy, School of Allied Health Sciences, Walailak University, Tha Sala, Nakhon Si Thammarat, Thailand

* nitita.do@wu.ac.th

**Data Availability Statement:** All relevant data are within the paper and its Supporting information file.

## Abstract

The COVID-19 global pandemic has had a socioeconomic effect, including many people suffering from stress and mental disorders. Health professional students are at risk of health issues as well when compared to their age-matched counterpart in the population. The present study aimed to find out the impact of COVID-19 awareness on the wellness of Thai health professional students. The awareness of COVID-19 and wellness among Thai health professional students, such as medical, physical therapy, nursing, pharmacy, and medical technology students were surveyed during the early "new normal" informing policy. The participants included 1,001 students, aged 17 to 25 years old, who responded to a Google form questionnaire set by request. The results showed that the prominent health risks among the Thai health professional students included sedentary behavior, obesity, and mental symptoms. A positive rating in attitude towards the COVID-19 epidemic was mostly observed. There was a negative influence on anxiety symptoms (standardized coefficient beta = -0.079, $p$-value = 0.012), but a positive impact on social well-being (standardized coefficient beta = 0.158, $p$-value < 0.001) and quality of life (standardized coefficient beta = 0.136, $p$-value < 0.001) among the students even when the situation was improving. To relieve the domino effect of the COVID-19 pandemic on students' wellness, an updated policy for enhancing awareness and providing updated information is continuously required. Improvements on self and situational awareness may help prevent health risk behaviors and promote health among Thai health professional students.

## Introduction

The Corona Virus Disease-2019 (COVID-19) pandemic has been declared as an international public health emergency [1, 2]. Thailand reported their 1st lab-confirmed case outside of China on 13 January 2020; herein, the patient departed from Wuhan, Hubei Province, China

**Funding:** Research grant came from the Center for Scientific and Technological Equipment, Walailak University, Thailand. This funding source had no role in the design of the study and did not have any role in collection, analysis, and interpretation of data or in the writing of the manuscript.

**Competing interests:** The authors declare that they have no conflict of interest.

[1]. Thereafter, the Thai government declared a state of emergency, where a curfew and lockdown measures went into effect. The COVID-19 crisis has widely caused detrimental ecological and social effects [3, 4]. Regarding personal wellness, wellness has been defined as a holistic integration of physical, mental, and spiritual well-being. It always includes striving for health [5]. Current studies have reported mental health problems during the COVID-19 pandemic [6–10].

Although the curfew in Thailand was lifted in July 2020, the state of emergency is still in effect [3]. Social distancing, self-isolation, and travel restrictions have caused job losses and increased the unemployment rate [3, 4]. The confirmed cases and deaths due to the COVID-19 widespread transmission are increasing [1]. These thereby have caused public stress among Thais [1, 3]. Together, it is necessary to quickly and continuously assess health problems related to emergencies among civilians, especially in vulnerable groups. This can provide an assessment of the impact and for policy implications. It is also needed for reviving the socioeconomic effects resulting from the crisis.

Health professional students when compared to their age-matched counterpart in the population usually have health risk behaviors due to facing multiple stressors. It was reported that half of them had at least one health risk behavior [11–14]. Wellness is significantly attributed to academic success and future career persistence. There should be an emphasis on studying it for improving health guidelines and counseling services, especially during an outbreak situation with socioeconomic effects [9–14].

Awareness varies along multiple dimensions, and it is defined as an intentional relationship between an organism and the item or situation in which it needs to be aware of [15–17]. Awareness is closely linked to the operation of the working memory [16, 17]. There has been reported evidence on the association between the awareness of chronic diseases and healthy lifestyles [18, 19]. In addition, it has been supported that to build more awareness may improve health through enhancing self-care [18]; however, there is very little research on awareness. The present study aimed to determine the impact of COVID-19 awareness on the wellness indicators of Thai health professional students during the situation. Sociodemographic characteristics, adherence to physical activity and exercise, mental symptoms, social well-being, and quality of life among students were revealed. It was hypothesized that awareness of COVID-19 may have an impact on the wellness indicators during the epidemic. If students have more awareness of COVID-19, their wellness during the global pandemic tends to be more achievable.

## Materials and methods

### Ethical approval and consent to participate

The study protocol in the present study received ethical approval from the Ethical Clearance Committee on Human Rights related to research involving human subjects, Walailak University, Thailand. The ethics code was 20-291-21. Informed consent for participation was obtained from all participants, in which they received the information sheet through an online platform.

### Participants

The present study was a cross-sectional analysis undertaken during September to October 2020. Sample size was calculated according to rules of thumb which was more than 74 (n > 50 + 8m (3 independent variables)) [20]. Convenient sampling in the present study recruited 1,001 Thai health professional students. This therefore avoided overfitting problems in regression models.

Research information was announced online via social online platforms (i.e., "Facebook" and "Line"). The inclusion criterion was undergraduate Thai health professional students, aged 17 to 25 years old. The exclusion criterion was students who had withdrawn from study or currently on internship. Year 6 medical, dental medical, veterinary medical, and pharmacy students were usually on their internship. There were no students who asked to decline from participation.

## Ambulatory assessment

Wellness among the Thai health professional students was surveyed via an ambulatory assessment. This is a computer-assisted methodology for studying people in their natural environment. Self-report or observational approach has been used for gathering data. These have been more ecologically valid than traditional field methodologies [21, 22]. Ambulatory assessment has been reported to minimize retrospective biases and to capture momentary ratings [21, 22]. Computerization has made the data collection more feasible, where participants using electronic devices typically show a higher rate of compliance at the time of the scheduled prompt [21]. Besides the benefits as reported by previous studies, ambulatory assessment also fits with the "new normal" policy.

In the present study, the self-report mode of ambulatory assessment was undertaken. Participants used their own devices to respond to the online questionnaires and tests. The Google form set included: 1) sociodemographic questions and rating scale on preference of learning under the "new normal", 2) Stroop test, 3) rating scale on awareness of COVID-19 and test of COVID-19, 4) physical activity questionnaire, 5) perceived stress scale, 6) mood disorder questionnaire, 7) depression symptom questionnaire, 8) anxiety scale, 9) multidimensional scale of perceived social support, and 10) quality of life questionnaire.

Research members provided the questionnaires and tests instructions. Then, participants were asked to complete the Google form, in which all of them finished within 30 minutes. The expected duration was 20 to 30 minutes, which was not over their concentration span. Although self-report was undertaken, participants could ask research members via meetings through the "Zoom" application if they were unclear about any questions or instructions during their response to the questionnaires and tests. However, no students requested additional instructions. All participants completely answered the questionnaires and tests.

## Questionnaires

The awareness scale for COVID-19 was modified from the scale represented in the study of Shuang-Jiang Zhou et al. (2020) [9]. The questionnaires included 3 rating scales, which are as follows: 1) "familiar with information about prevention and control of COVID-19" indicating knowledge of COVID-19, 2) "take response to the prevention and control measures against COVID-19 to avoid infection" representing practice following infection control measures, and 3) "attitudes toward the projection of COVID-19 trends" determining attitude towards the COVID-19 pandemic. Herein, 1 was very unfamiliar, very inconsistent, or very pessimistic; and 10 was very familiar, very consistent, or very optimistic. To confirm the self-rating scale on awareness, 19 additional true-false questions on COVID-19 in regard to knowledge and practice were asked. These questions were modified from the questionnaire in the study of Bao-Liang Zhong et al. (2020) [7] and from the World Health Organization (WHO), 2020 [1]. Besides the awareness rating scale, preference on learning under the "new normal" was scored from 1 (very unlikely) to 10 (very likely) as well. Cognitive function was also determined in all participants through the online Stroop test. The test score presented the total number of

correct answers, where higher scores reflected better performance and less interference on reading ability [23].

Wellness of Thai health professional students were determined in the present study by using the International Physical Activity Questionnaire Short Form (IPAQ-SF), Perceived Stress Scale (PSS-10), Mood Disorder Questionnaire (MDQ), 7-item anxiety scale (GAD-7), Patient Health Questionnaire (PHQ-9), Multidimensional Scale of Perceived Social Support (MSPSS), and the World Health Organization Quality of Life (WHOQOL-BREF). IPAQ-SF was used to assess physical activity level and walking Metabolic Equivalent (MET) in minutes per week. The obtained-total MET minutes per week was classified as sedentary behavior, moderate physical activity, or high physical activity. Scores of PSS-10 were classified as low stress (0 to 13), moderate stress (14 to 26), or high stress (27 to 40). MDQ was used to screen bipolar disorder, where a total score equal to or higher than 7 was identified as potentially suffering from bipolar disorder. Scores of GAD-7 were classified as normal anxiety (0 to 9), moderate anxiety (10 to 14), or high anxiety (15 to 21). PHQ-9 was used to determine depression symptoms, in which a score less than 7 was no symptoms, 7 to 12 was mild, 13 to 17 was moderate, and equal to or more than 18 was severe. Social well-being was determined through the rating of the MSPSS, where higher scores indicated a greater perceived social support. Quality of life was considered as poor, moderate, and good; herein, 26 to 60 from WHOQOL-BREF indicated poor, 61 to 95 was moderate, and 96 to 130 was good.

All questionnaires in the present study were represented in the Thai language through a Thai version or translation provided. Most of these instruments in Thai version have been used among Thais, including the online Stroop test, IPAQ-SF, PSS-10, MDQ, GAD-7, PHQ-9, MSPSS, and WHOQOL-BREF. The validity and reliability of the questionnaires have been reported in previous studies [24–31]. Literature reviews have supported the awareness scale for COVID-19, the 19 additional true-false questions on COVID-19, and a created-numerical rating scale for determining preference on learning under the "new normal" as first used among Thais. It has been supported that a rating scale is valid if it measures what it is intended to measure in the specific study [32].

## Statistical analysis

Data analysis was conducted by using SPSS V26. The correlation between the sociodemographic characteristics with the awareness of COVID-19 and wellness variables was determined by the Spearman's correlation test. A regression analysis was performed to determine the impact of COVID-19 awareness (i.e., knowledge, practice, and attitude) and sociodemographic characteristics (i.e., monthly income and accumulated grade point average (GPAX)) on anxiety, social well-being, and quality of life. The statistical significance level was $p$-value $\leq 0.05$.

## Results

Cross-sectional survey data shown in the present study was obtained from 1,001 Thai health professional students, aged 17 to 25 years old, through their responses on the Google form questionnaires set. The sociodemographic data showed that there was a higher proportion of female health professional students answering the Google form (81%). Among the Thai health professional students in 14 programs, approximately 71% of the responses came from Year 1 and 2 students. The number of responses decreased by years of study (i.e., Year 3, Year 4, and Year 5, respectively). Most of the health professional students had a GPAX score of more than 2.5, and only 6% of them had a lower GPAX score (Table 1). The main source of income of most of the Thai health professional students based on their response was income received

**Table 1. Sociodemographic characteristics of the participants.**

| Participant characteristics (n = 1,001) | Number (percentage) |
|---|---|
| Sex | |
| Female | 808 (80.72) |
| Male | 193 (19.28) |
| Program of study | |
| Dental Medicine | 14 (1.40) |
| Emergency Medical Operation | 2 (0.20) |
| Medical Technology | 163 (16.28) |
| Medicine | 73 (7.29) |
| Nursing | 148 (14.79) |
| Occupational Therapy | 11 (1.10) |
| Pharmacy | 200 (19.98) |
| Physical Therapy | 263 (26.27) |
| Public Health | 60 (5.99) |
| Radiological Technology | 15 (1.50) |
| Science and Health Technology | 6 (0.60) |
| Sport Science and Exercise | 17 (1.70) |
| Thai Traditional Medicine | 19 (1.90) |
| Veterinary Medicine | 10 (1.00) |
| Year of study | |
| Year 1 | 341 (34.07) |
| Year 2 | 370 (36.96) |
| Year 3 | 154 (15.38) |
| Year 4 | 124 (12.39) |
| Year 5 | 12 (1.20) |
| Accumulated grade point average (GPAX) | |
| < 2.50 | 62 (6.19) |
| 2.50–3.00 | 237 (23.68) |
| 3.01–3.50 | 395 (39.46) |
| 3.51–4.00 | 307 (30.67) |
| Main source of income | |
| Parents or guardian | 963 (96.20) |
| Scholarship | 32 (3.20) |
| Job | 6 (0.60) |
| Income [Thai Bath (United States Dollar)/month] | |
| < 5,000 (< 159.08) | 513 (51.25) |
| 5,000–10,000 (159.08–318.17) | 445 (44.46) |
| 10,001–15,000 (318.20–477.25) | 25 (2.50) |
| > 15,000 (> 477.25) | 18 (1.79) |
| Feeling enough income | |
| Enough | 775 (77.42) |
| Not enough | 226 (22.58) |
| Body Mass Index (kg/m$^2$) | |
| Underweight (< 18.5) | 178 (17.78) |
| Normal (18.5–22.9) | 500 (49.95) |
| Overweight (23–24.9) | 140 (13.99) |
| Obese I (25–29.9) | 124 (12.39) |
| Obese II ($\geq$ 30) | 59 (5.89) |

(*Continued*)

**Table 1.** (Continued)

| Participant characteristics (n = 1,001) | Number (percentage) |
|---|---|
| Exercise behavior and adherence during past 3 months | |
| Not exercising | 334 (33.36) |
| Exercising < 3 days/week | 505 (50.45) |
| Exercising ≥ 3 days/week | 162 (16.19) |
| Smoking behavior | |
| Smoking | 117 (11.69) |
| Not smoking | 884 (88.31) |
| Alcoholic drinking behavior | |
| Drinking alcoholic beverages | 286 (28.57) |
| Not drinking alcoholic beverages | 715 (71.43) |

GPAX represented in the present study was self-reported by the students. Thai students know their current GPAX from login the university's website. Conversion from Thai baht to United States Dollar was based on the exchange rate for April 23, 2021, as provided by Morningstar for Currency and Coinbase for Cryptocurrency.

from their parents or guardian (96%). The total income of the students mostly ranged from 10,000 baht to less than 5,000 baht (96%), and around 77% of the students felt that it was enough. Other main sources of income included scholarships and jobs (Table 1). Regarding health, their height and weight within the past month was self-reported in order to determine their body mass index. It was found that about half of the students had normal body mass index, where 32% were overweight to obese, and 18% were underweight. According to the American College of Sports Medicine (ACSM)'s Guidelines [33], it indicated that approximately 84% of these students met the conditions for sedentary behavior. They did not participate in moderate physical activity (that resulted in perspiration) at least 3 days/week for 3 months. Smoking and/or drinking alcohol beverages were reported in 12% to 29% of the health professional students (Table 1).

Awareness of COVID-19 was determined among the Thai health professional students during the academic reopening and early "new normal" informing policy. It was reported that the Thai health professional students knew and practiced prevention and control measures in regard to COVID-19, and they also had a positive attitude towards the projection of the COVID-19 pandemic (Table 2). Consistent with the collected data on COVID-19 knowledge and practice, the average score of the COVID-19 knowledge and practice test (modified from the study of Bao-Liang Zhong et al., 2020 and from the WHO, 2020) was also high (85%)

**Table 2. Self-report scales for COVID-19 awareness and preference of learning under the "new normal", and the Stroop test score of the Thai health professional students.**

| Determinants (n = 1,001) | Median (min–max) | Mode (percentage) |
|---|---|---|
| Self-report scales | | |
| COVID-19 awareness (1–30) | 23 (9–30) | 25 (10.30) |
| COVID-19 knowledge (1–10) | 8 (1–10) | 8 (28.30) |
| COVID-19 protection practice (1–10) | 8 (1–10) | 9 (23.10) |
| Attitude towards COVID-19 pandemic (1–10) | 8 (1–10) | 8 (24.60) |
| Preference of learning under the "new normal" (1–10) | 6 (1–10) | 5 (16.60) |
| Stroop test | | |
| Score (1–20) | 19 (1–20) | 19 (40.30) |

(S1 Table). In contrast, the preference for online learning was just at midpoint (Table 2). Since awareness has been reported to be related with cognitive function, a cognitive test score was also determined in the present study, where 1,001 health professional students were requested to take the Stroop test. The average score for this test was approximately 90% (Table 2).

To observe personal wellness with a holistic approach, physical activity, stress, mood, anxiety symptoms, depression symptoms, social well-being, and quality of life among the Thai health professional students were determined in the present study by using IPAQ-SF, PSS-10, MDQ, GAD-7, PHQ-9, MSPSS, and WHOQOL-BREF, respectively (Table 3). In agreement with ACSM's recommendation, the IPAQ-SF data analysis also revealed that most of the Thai health professional students spent their time in sedentary behavior (Table 3). Moderate stress and normal mood were mostly reported. Regarding symptoms of anxiety and depression, about 82% of the Thai health professional students had normal anxiety symptoms and around half of them had depression symptoms with 19% at mild, 10% at moderate, and 26% at severe

**Table 3. Physical activity, stress, mood, symptoms of anxiety and depression, social well-being, and quality of life among the Thai health professional students.**

| Determinants (n = 1,001) | Number (percentage) | Median (min–max) |
|---|---|---|
| Physical activity (IPAQ-SF) | | |
| Sedentary | 747 (74.63) | - |
| Moderate | 166 (16.58) | - |
| High | 88 (8.79) | - |
| Walking MET (minutes/week) | - | 40 (0–280) |
| Stress (PSS-10) | | |
| Low | 239 (23.88) | - |
| Moderate | 725 (72.43) | - |
| High | 37 (3.69) | - |
| Mood (MDQ) | | |
| Normal | 962 (96.10) | - |
| Bipolar | 39 (3.90) | - |
| Anxiety (GAD-7) | | |
| Normal | 818 (81.72) | - |
| Moderate | 134 (13.39) | - |
| High | 49 (4.89) | - |
| Depression (PHQ-9) | | |
| No symptoms | 447 (44.66) | - |
| Mild | 193 (19.28) | - |
| Moderate | 101 (10.09) | - |
| Severe | 260 (25.97) | - |
| Social well-being (MSPSS) | - | 70 (12–84) |
| Quality of life (WHOQOL-BREF) | | |
| Poor | 49 (4.90) | - |
| Moderate | 519 (51.85) | - |
| Good | 433 (43.25) | - |

IPAQ-SF is International Physical Activity Questionnaire-Short Form, Walking MET is Walking Metabolic Equivalent, PSS-10 is Perceived Stress Scale, MDQ is Mood Disorder Questionnaire, GAD-7 is 7-item anxiety scale, PHQ-9 is Patient Health Questionnaire, MSPSS is Multidimensional Scale of Perceived Social Support, and WHOQOL-BREF is the World Health Organization Quality of Life.

(Table 3). Nevertheless, normal social well-being and moderate quality of life were mostly reported by the students (Table 3).

The association between COVID-19 awareness and physical activity, stress, mood, anxiety symptoms, depression symptoms, social well-being, and quality of life was determined. Regarding physical activity, it was found that COVID-19 awareness was associated with walking MET (Table 4). Regarding mental health, COVID-19 awareness was positively associated with mood but negatively related with anxiety. There were no significant relationships between COVID-19 awareness with stress and symptoms of depression (Table 4). Social well-being and quality of life were positively related with COVID-19 awareness (Table 4).

All three aspects of COVID-19 awareness, including knowledge, practice, and attitude were separated in order to reveal their impacts on mood, anxiety symptoms, social well-being, and quality of life among the Thai health professional students when compared to the effects related to the sociodemographic characteristics, i.e., income and GPAX (Table 5 and S2 Table). A regression analysis indicated that there were no observed factors influencing mood, but the attitude towards the COVID-19 pandemic and total income did influence anxiety symptoms among the Thai health professional students. The attitude rather than income had a higher effect on anxiety symptoms (Table 5). The impacts of knowledge, practice, and GPAX on anxiety symptoms were not found. Social well-being and quality of life as reported during the early "new normal" informing policy were solely affected by the attitude and knowledge of COVID-19 awareness (Table 5 and S2 Table).

**Table 4. Association between COVID-19 awareness and physical activity, stress, mood, symptoms of anxiety and depression, social well-being, and quality of life.**

| | COVID-19 awareness (n = 1,001) |
|---|---|
| Walking MET minutes a week | |
| Correlation ($r$) | 0.052* |
| $p$-value | 0.049 |
| Stress | |
| Correlation ($r$) | 0.400 |
| $p$-value | 0.101 |
| Mood | |
| Correlation ($r$) | 0.052* |
| $p$-value | 0.049 |
| Anxiety symptoms | |
| Correlation ($r$) | -0.061* |
| $p$-value | 0.028 |
| Depression symptoms | |
| Correlation ($r$) | -0.027 |
| $p$-value | 0.201 |
| Social well-being | |
| Correlation ($r$) | 0.183*** |
| $p$-value | < 0.001 |
| Quality of life | |
| Correlation ($r$) | 0.087* |
| $p$-value | 0.003 |

* $p$-value $\leq$ 0.05;

*** $p$-value $\leq$ 0.001

**Table 5. Attitude towards the COVID-19 pandemic and its influence on anxiety, social well-being, and quality of life among Thai health professional students during the early "new normal" informing policy.**

| Variables (n = 1,001) | R square | Coefficients Std. Error | Standardized Coefficient Beta | t | p-value |
|---|---|---|---|---|---|
| Dependent variable: Mood | 0.003 | | | | |
| Attitude | | 0.069 | 0.051 | 1.608 | 0.108 |
| Income | | 0.000 | 0.015 | 0.478 | 0.633 |
| GPAX | | 0.200 | 0.006 | 0.178 | 0.858 |
| Dependent variable: Anxiety Symptoms | 0.011 | | | | |
| Attitude | | 0.090 | -0.079 | -2.523 | 0.012* |
| Income | | 0.000 | 0.064 | 2.044 | 0.041* |
| GPAX | | 0.259 | 0.022 | 0.700 | 0.484 |
| Dependent variable: Social well-being | 0.026 | | | | |
| Attitude | | 0.249 | 0.158 | 5.053 | < 0.001*** |
| Income | | 0.000 | 0.025 | 0.787 | 0.431 |
| GPAX | | 0.721 | -0.020 | -0.655 | 0.513 |
| Dependent variable: Quality of life | 0.022 | | | | |
| Attitude | | 0.283 | 0.136 | 4.331 | < 0.001*** |
| Income | | 0.000 | 0.051 | 1.637 | 0.102 |
| GPAX | | 0.820 | -0.022 | -0.688 | 0.491 |

* p-value ≤ 0.05;

*** p-value ≤ 0.001

## Discussion

An ambulatory assessment of wellness among Thai health professional students during the academic reopening and early "new normal" informing policy revealed that the proportion of normal to abnormal body mass index was about 50 to 50. The majority of the students met the conditions for sedentary behavior. Smoking and/or drinking alcoholic beverages were reported in about one-fourth of the students. It was found that most of the Thai health professional students had normal anxiety symptoms. Around half of them had depression symptoms. The Thai health professional students still perceived social well-being during this study. However, moderate quality of life was determined in half of the students; and some of them had poor quality of life. The reported health risks among these Thai health professional students were similar to those reported in previous studies among adolescents in which sedentary behavior, obesity, and mental health problems had been prominent [34–37]. The present study strongly supports evidence-informed policies for preventing health risk behaviors and promoting wellness among Thai youth, where factors resulting in health risk behaviors among them should be clarified.

Regarding mental health problems, the finding of a high prevalence of mental symptoms was consistent with a cross-sectional study by Shuang-Jiang Zhou in 2020, where depression and anxiety symptoms in 8,079 Chinese adolescents aged 12 to 18 years old were determined during the COVID-19 epidemic outbreak [9]. A domino effect of COVID-19 on mental health has been suggested [6–10]; herein, the Thai health professional students also were affected by COVID-19 even when the situation was improving. Students with mental symptoms are at risk of mental health disorders in adulthood if they remain undetected or are not treated appropriately [9, 22]. It has been reported that anxiety and depression symptoms in health professional students adversely affected the quality of patient care, patient safety, and

professionalism in which these effects extended to society [11–14]. Thus, these students' mental health should be a prioritized concern by their respective universities [11].

The awareness level of COVID-19 as shown in the present study was confirmed by the scores from the COVID-19 knowledge and practice test and the Stroop test. The students rated their COVID-19 awareness at a high level. This supported Thailand's successful response to the first outbreak by communicating with this generation through social media [38]. Even with the high level of COVID-19 awareness that was observed among the students, their rating for preference of learning during this situation by contrast was just at midpoint. This implied that it might be difficult to adjust their learning behavior under the "new normal." The students might feel uncomfortable and mildly anxious, which might be related with the reported mental symptoms.

In agreement with a recent study by Shuang-Jiang Zhou and colleagues, it was found that the awareness of COVID-19 was related with psychological health problems [9]. COVID-19 awareness was positively associated with mood but negatively related with anxiety symptoms. There were no relationships between COVID-19 awareness with stress and depression symptoms. The association between the awareness of COVID-19 with mood and anxiety symptoms implied that the students could control their emotions. Stress and depression symptoms may depend on other chronic effects, which were not related to COVID-19 awareness. Psychological trauma, adverse childhood experiences, genetic predisposition, and personality traits have been found to be risk factors of psychological disorders [39]. However, the causes of most mental disorders are not fully understood; and a variety of biological, psychological, and environmental factors usually combined contribute to the development of mental disorders [39].

Besides mental health, the association between the awareness of COVID-19 and walking MET was observed in the present study. Although the relationship between physical activity level and awareness of chronic diseases has been previously reported [18], most of the students usually have low physical activity and did not exercise. This association, thus, suggested poor self-care health behaviors among the students. The present study supports the improvement of COVID-19 awareness among the students because there were correlations between the awareness of COVID-19 with social well-being and quality of life.

A regression analysis indicated that the attitude towards the COVID-19 pandemic, rather than total income, had a higher effect on anxiety symptoms in the Thai health professional students. Their social well-being and quality of life were solely affected by their attitude and knowledge of COVID-19. Informed and updated information on COVID-19, therefore, is continuously required for supporting wellness among Thai health professional students. The present study also supports the impact of awareness on health status, where promoting awareness may be the goal of preventing health risks in adolescents and young adults.

Although wellness was assessed with a holistic approach in the participants' natural environment; the effects of COVID-19 awareness represented in the present study were based on self-reports collected from questionnaires and tests. Other factors may have influenced wellness among the Thai health professional students. Body mass index as reported in the present study was determined only through self-reports, which might have affected the accuracy of the results. The exact compliance rate and number of declined participations could not be determined through the self-reports and convenient sampling.

## Conclusions

The present study indicated that the awareness of COVID-19 had an influence on the symptoms of anxiety, social well-being, and quality of life among Thai health professional students even when the situation was improving. Updated policy is continuously required for

enhancing positive attitude and providing updated information. Improvements on self and situational awareness may help prevent health risk behaviors and promote health among Thai health professional students.

## Supporting information

**S1 Table. The score of the COVID-19 knowledge and practice tests.**
(DOCX)

**S2 Table. Knowledge of COVID-19 influences on social well-being and quality of life among the Thai health professional students during the early "new normal" informing policy.**
(DOCX)

**S1 Raw data. Sociodemographic characteristics of the participants: Sex, program of study, year of study, GPAX, main source of income, income, feeling enough income, BMI, exercise behavior, smoking behavior, and alcoholic drinking behavior.**
(PDF)

**S2 Raw data. Self-report scales for COVID-19 awareness and preference of learning under the "new normal", and the Stroop test score of the Thai health professional students_S1 The score of the COVID-19 knowledge and practice tests.**
(PDF)

**S3 Raw data. Physical activity, stress, mood, symptoms of anxiety and depression, social well-being, and quality of life among the Thai health professional students.**
(PDF)

**S4 Raw data. Association between COVID-19 awareness and physical activity, stress, mood, symptoms of anxiety and depression, social well-being, and quality of life: Stress, mood, anxiety symptoms, depression symptoms, and quality of life_T5 Attitude towards the COVID-19 pandemic and its influence on anxiety, social well-being, and quality of life among Thai health professional students during the early "new normal" informing policy: GPAX, income, anxiety symptoms, and quality of life_S2 Knowledge of COVID-19 influences on social well-being and quality of life among the Thai health professional students during the early "new normal" informing policy: GPAX, income, and quality of life.**
(PDF)

## Acknowledgments

The authors would like to thank all participants for taking the time to complete this survey. We would like to express our gratitude to Prof. Dr. Narattaphol Charoenphandhu, MD, PhD for his kind support. We would also like to thank Mr. David C. Chang for English manuscript editing.

## Author Contributions

**Conceptualization:** Krueakaew Tiaprapong, Nitita Piya-amornphan.

**Data curation:** Krueakaew Tiaprapong, Achiraporn Sirikul, Chamawee Krajangmek, Namfon Duangthongkul, Nichaya Pandam, Nitita Piya-amornphan.

**Formal analysis:** Krueakaew Tiaprapong, Achiraporn Sirikul, Chamawee Krajangmek, Namfon Duangthongkul, Nichaya Pandam, Nitita Piya-amornphan.

**Investigation:** Achiraporn Sirikul, Chamawee Krajangmek, Namfon Duangthongkul, Nichaya Pandam.

**Project administration:** Nitita Piya-amornphan.

**Supervision:** Nitita Piya-amornphan.

**Writing – original draft:** Nitita Piya-amornphan.

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
