## [Decision Letter · Decision Letter 0]

13 Apr 2021

PONE-D-21-00065

Awareness of COVID-19 influences on the wellness of Thai health professional students: An ambulatory assessment during the early "new normal" informing policy

PLOS ONE

Dear Dr. Piya-amornphan,

Thank you for submitting your manuscript to PLOS ONE. After careful consideration, we feel that it has merit but does not fully meet PLOS ONE’s publication criteria as it currently stands. Therefore, we invite you to submit a revised version of the manuscript that addresses the points raised during the review process.

The reviewers have recommended major revisions to your manuscript. Therefore, I invite you to respond to the reviewers comments and revise your manuscript. The comments of the reviewer are included at the end of this letter.

We look forward to receiving your revised manuscript.

Kind regards,

Enkeleint A. Mechili

Academic Editor

PLOS ONE

Journal Requirements:

3. PLOS ONE has specific requirements for studies using personal data from third-party sources, including social media, blogs, other internet sources, and phone companies (https://journals.plos.org/plosone/s/submission-guidelines#loc-personal-data-from-third-party-sources). These requirements include confirming data are collected and used in accordance with the company or website’s Terms and Conditions, obtaining appropriate ethics or data protection body review, and ensuring appropriate consent from individuals whose data are used in research. In this case, please ensure that your Ethics statement is in compliance with guidelines, and that you have complied with the company's (e.g., Facebook) Terms and Conditions, with appropriate permissions.

*Please include additional information regarding the survey or questionnaire used in the study and ensure that you have provided sufficient details that others could replicate the analyses. For instance, if you developed a questionnaire as part of this study and it is not under a copyright more restrictive than CC-BY, please include a copy, in both the original language and English, as Supporting Information.

*Please note that PLOS ONE does not copy edit accepted manuscripts (https://journals.plos.org/plosone/s/criteria-for-publication#loc-5). To that effect, please ensure that your submission is free of typos and grammatical errors.

Additional Editor Comments (if provided):

Reviewers' comments:

Reviewer's Responses to Questions

**Comments to the Author**

1. Is the manuscript technically sound, and do the data support the conclusions?

Reviewer #1: Yes

Reviewer #2: No

2. Has the statistical analysis been performed appropriately and rigorously? 

Reviewer #1: Yes

Reviewer #2: I Don't Know

3. Have the authors made all data underlying the findings in their manuscript fully available?

Reviewer #1: Yes

Reviewer #2: Yes

4. Is the manuscript presented in an intelligible fashion and written in standard English?

Reviewer #1: Yes

Reviewer #2: No

5. Review Comments to the Author

Reviewer #1: This is a very interesting article that aimed to assess wellness and the awareness of COVID-19 among Thai health professional students.

Abstract.

- It is not very clear what is ambulatory assessment.

- "The questionnaires achieved 100% compliance among the participants" Please clarify.

Introduction

- Page 3 and 4 rows 57-58. Please add reference.

- Page 4 row 66. It is not clear what authors mean

- Rows 70-89 please shorten this part and be more specific. Is very broad and reader is loosing the concept.

- Rows 90-97. Sentences are very long. Please split them.

- You are presenting the aim of the current article and are putting references (row 100). Don't know why. Please explain.

- You are presenting twice the aim of the study.

- Introduction part should be more concise and clear. I'm suggesting re-writing this part.

Methods

- There were no students who declined participation (page 7 rows 126-127). How you are sure about this when the information was sent by Facebook and Line?

- Have been used these instruments in the past in Thai?

- Did you check validity and reliability of the questionnaires used?

- What do you mean with the terms ambulatory assessment. It is not clear.

- What is GPAX.

Results

- Please mention the income also in US dollars

- How did yuou assess BMI?

- Page 13, row 230 add reference

- As Medicine normally is 6 years. Did you have participants from this year?

- Rows 242-247 please include them in methods part;

Discussion

- Please do not repeat numbers from the results part

- Which are the limitations of this study. Add a section about strengths and limitations

Reviewer #2: This is an interesting topic. Please see my feedback for further improvement.

The title of the study does not match with the aims of the study written in the abstract and introduction section.

The second aim, i.e. “The impact of COVID 19 awareness on improving health during the situation was also clarified” needs more clarity. How did you assess the improving health of the participants? I suggest you to write the hypothesis of the study for better understanding the study.

‘Wellness’ is one of the important variables in your study. Wellness is a state of complete physical, mental, and social well-being. But you have highlighted only the mental health of the students in the introduction section. Your focus should be shifted to wellness rather than mental health.

It is not clear why the authors have written in page 6 lines 97 to 103 as “ The present study aimed to reveal health risk behaviours………characteristics were surveyed” Please read the paragraph and correct it.

Method section needs improvement. Write the design of the study. How did you determine the sample size?

You have defined, Ambulatory assessment as a method to study people in their daily life, which consists of a wide range of assessments including momentary self-report, observational assessment, and physiological assessment. One of its’ essential features is observational assessment. How did you meet this feature via online mode?

You have used a number of questionnaires to assess the wellness of the Thai Health Professionals. However, information regarding the scoring directions and Cronbach reliability coefficient scores have not been given. Also, proper direction is required for the Stroop test.

Page 11, lines 202 to 205: In statistical analysis section, you have mentioned Anxiety, social well-being and quality of life as socio-demographic characteristics, whereas it is different in table 1. (There is confusion in bracket, please, correct it)

Page 12, line 217: Write programmes in place of curriculum

Page 12, line 220: Mention the full form of GPAX while writing first time in the article. Your study also covered year 1 students. Did they get GPAX during the study? You also need to write the full form of other abbreviations such as ‘Walking MET’ (Page 19, line 291).

Page 13, lines 230-232: What is the relevant of writing “According to the American College of Sports Medicine (ACSM)’s Guidelines, it indicated that approximately 84% of the students met the conditions for sedentary behaviour” in the result section.

Page 15, lines 240 to 247: Description of the questionnaire has already been given in the methodology section. There is no reason to repeat the same in the result section.

Page 16, lines 255 to 256: What does ‘moderate ‘ mean in terms of learning preference? It should be reflected in the methodology section.

Page 17, line 277-278: What is high-normal anxiety? Either, it should be severe or normal.

Page 19, line 294: It should be no significant relationship

In order to measure the stress level of the students, you have used Perceived Stress. You have written ‘emotional stress’ in some places, which creates confusion. So, write ‘stress’ in place of ‘emotional stress’.

Table 5 need more clarity. You have stated, “All three aspects of COVID-19 awareness including knowledge, practice, and attitude were separated in order to reveal their impacts on mood, anxiety symptoms, social well-being, and quality of life among the Thai health professional students when compared to the effects related to the sociodemographic characteristics, i.e. income and GPAX score.” You have 3 aspect of COVID-19 awareness. However, in table 5 only ‘attitude’ has been reported.

6. PLOS authors have the option to publish the peer review history of their article (what does this mean?). If published, this will include your full peer review and any attached files.

Reviewer #1: **Yes: **Enkeleint A. Mechili

Reviewer #2: **Yes: **Pradeep Kumar Sahu

---

## [Author Response · Author response to Decision Letter 0]

5 May 2021

Response to reviewer comments for manuscript

PONE-D-21-00065

"Awareness of COVID-19 influences on the wellness of Thai health professional students: An ambulatory assessment during the early "new normal" informing policy"

We would like to thank you the reviewers for your valuable comments and suggestions to help us improve the manuscript. We have made revisions throughout the manuscript according to the reviewers’ comments and suggestions. We put forth an effort to rewrite the manuscript based on all the comments and suggestions provided. Our responses to the reviewers’ comments are listed in the following table.

Reviewer 1

Abstract:

- It is not very clear what is ambulatory assessment. 

We have made revisions throughout the abstract to refer to the appropriate presentation. We have clarified the term “ambulatory assessment” and have added additional information to explain this term in the methods section. Please see Page 7, lines 121-133.

- "The questionnaires achieved 100% compliance among the participants" Please clarify. 

We provided meetings through the “Zoom” application if participants were unclear about any questions or instructions during their response to the questionnaires and tests. “The questionnaires achieved 100% compliance among the participants” thus came from assumption. We have removed this sentence from the abstract and addressed the limitation of the method on Page 27, lines 413-421.

Introduction:

- Page 3 and 4 rows 57-58. Please add reference. 

We have added references supporting the information.

- Page 4 row 66. It is not clear what authors mean 

We have removed “only symptomatic treatment is the best possible chance for survival” to make the paragraph more concise.

- Rows 70-89 please shorten this part and be more specific. Is very broad and reader is loosing the concept. We have made revisions throughout the introduction section, and added additional information emphasizing the study concept. Please see the introduction section Pages 3-6.

- Rows 90-97. Sentences are very long. Please split them. 

We have revised these sentences. It has been changed to: “Health professional students when compared to their age-matched counterpart in the population usually have health risk behaviors due to facing multiple stressors. It was reported that half of them had at least one health risk behavior. Wellness is significantly attributed to academic success and future career persistence. There should be an emphasis on studying it for improving health guidelines and counseling services, especially during an outbreak situation with socioeconomic effects.”

- You are presenting the aim of the current article and are putting references (row 100). Don't know why. Please explain. 

We apologize for the typographical error. We have moved that reference to the sentences: “Wellness is significantly attributed to academic success and future career persistence. There should be an emphasis on studying it for improving health guidelines and counseling services, especially during an outbreak situation with socioeconomic effects.” 

- You are presenting twice the aim of the study. 

We have rewritten the objectives for clarity. Please see Page 5, lines 88-96.

- Introduction part should be more concise and clear. I'm suggesting re-writing this part. 

We have made revisions throughout the introduction section, and added additional information emphasizing the study concept as suggested. Please see the introduction section Pages 3-6.

Methods:

- There were no students who declined participation (page 7 rows 126-127). How you are sure about this when the information was sent by Facebook and Line? 

There were no students who asked to decline from participation during the “Zoom” meeting provided for facilitating their responses; therefore, we have changed this sentence to: “There were no students who asked to decline from participation.” for clarity. We also have addressed this as a limitation on Page 27, lines 413-421.

- Have been used these instruments in the past in Thai? 

Most of these instruments in the present study have been used among Thais, including the online Stroop test, IPAQ-SF, PSS-10, MDQ, GAD-7, PHQ-9, multidimensional scale of perceived social support, and WHOQOL-BREF. The validity and reliability of these instruments have been reported in previous studies. Literature reviews have supported the awareness scale for COVID-19 modified from the scale represented in the study of Shuang-Jiang Zhou et al. (2020) and the 19 additional true-false questions on COVID-19 modified from the questionnaire in the study of Bao-Liang Zhong et al. (2020) and from the WHO, 2020 as first used among Thais. We created a rating scale to determine preference on learning under the “new normal” where a numerical rating scale was used; herein, 1 was “very unlikely” and 10 was “very likely.” This scale was first used among Thais. We have added more text to clarify the questionnaires subsection. Please see Pages 9-12.

- Did you check validity and reliability of the questionnaires used? 

The validity and reliability of the online Stroop test, IPAQ-SF, PSS-10, MDQ, GAD-7, PHQ-9, MSPSS, and WHOQOL-BREF have been reported in previous studies. We have added references for this information. Literature reviews did not mention about the validity and reliability of the true-false questions on COVID-19; however, it has been supported that a rating scale is valid if it measures what it is intended to measure in the specific study. We also address this concern on Page 11, lines 199-210.

- What do you mean with the terms ambulatory assessment. It is not clear. 

We used the term “ambulatory assessment” to define a computer-assisted methodology for studying people in their natural environment. Herein, self-reports have been used for gathering data. Students used their own devices to respond to the online questionnaires and tests. We have added text to clarify this term in the ambulatory assessment subsection. Please see Page 7, lines 121-153.

- What is GPAX. 

We have added “accumulated grade point average” as the full name of “GPAX” in the statistical analysis subsection. We have used “GPAX” according to the questionnaire, GPAX represented in the present study was self-reported by the students. Thai students know their current GPAX from login the university’s website. We have represented this information by adding footnote in Table 1.

Results:

- Please mention the income also in US dollars 

We have converted Thai baht to US dollars based on the current exchange rate and have added this information in Table 1.

- How did you assess BMI? 

BMI was calculated from the obtained data on height and weight through the Google form. We asked participants to provide updated information, which was not past more than a month. We have added this information for clarity on Page 13, lines 239-240. We also addressed the limitation of the method on Page 27, lines 413-421.

- Page 13, row 230 add reference 

We have added the reference for the American College of Sports Medicine (ACSM)’s Guidelines.

- As Medicine normally is 6 years. Did you have participants from this year? 

After rechecking, we did not observe data from Year 6 students. This may be the result that Year 6 medical, dental medical, veterinary medical, and pharmacy students were usually on their internship, which was an exclusion criterion for the present study. We have added this information on Page 7, lines 117-119.

- Rows 242-247 please include them in methods part; 

We have removed: “The questionnaires included questions with 3 rating scales that were modified from ... Herein, 1 was minimum and 10 was maximum.” from the results section as suggested.

Discussion:

- Please do not repeat numbers from the results part 

We have corrected this as suggested.

- Which are the limitations of this study. Add a section about strengths and limitations 

We have added text to clarify the strengths and limitations of the present study in the discussion section as: “Although wellness was assessed with a holistic approach in the participants’ natural environment, the effects of COVID-19 awareness represented in the present study were based on self-reports collected from questionnaires and tests. Other factors may have influenced wellness among the Thai health professional students. Body mass index as reported in the present study was obtained from body height and weight through self-report, which might affect the accuracy of the results. The exact compliance rate and number of declined participations could not be determined through the self-reports and convenient sampling.”

Reviewer 2

- The title of the study does not match with the aims of the study written in the abstract and introduction section. 

We have revised the aims of the study presented in the abstract and introduction section to match with the title as suggested. Please see Page 2, lines 24-26 and Page 5, lines 88-90.

- The second aim, i.e. “The impact of COVID 19 awareness on improving health during the situation was also clarified” needs more clarity. How did you assess the improving health of the participants? I suggest you to write the hypothesis of the study for better understanding the study. 

We have revised the aims of the study presented in the introduction section for clarity and conciseness. We have changed that sentence to: “The present study aimed to determine the impact of COVID-19 awareness on the wellness indicators of Thai health professional students during the situation.” We also have added the hypothesis as: “It was hypothesized that awareness of COVID-19 may have an impact on the wellness indicators during the epidemic. If students have more awareness of COVID-19, their wellness during the global pandemic tends to be more achievable.” as suggested. Please see on Page 5, lines 88-96.

- ‘Wellness’ is one of the important variables in your study. Wellness is a state of complete physical, mental, and social well-being. But you have highlighted only the mental health of the students in the introduction section. Your focus should be shifted to wellness rather than mental health. 

We have made revisions throughout the introduction section, and added additional information emphasizing the study concept. We also have added text and reference to highlight ‘wellness’ as: “Regarding personal wellness, wellness has been defined as a holistic integration of physical, mental, and spiritual well-being. It always includes striving for health.” Please see the introduction section Pages 3-6.

- It is not clear why the authors have written in page 6 lines 97 to 103 as “ The present study aimed to reveal health risk behaviours………characteristics were surveyed” Please read the paragraph and correct it. 

We have rewritten the objectives presented in the introduction section for clarity and conciseness as suggested. These sentences have been changed to: “The present study aimed to determine the impact of COVID-19 awareness on the wellness indicators of Thai health professional students during the situation. Sociodemographic characteristics, adherence to physical activity and exercise, mental symptoms, social well-being, and quality of life among students during this difficult situation were revealed.”

- Method section needs improvement. Write the design of the study. How did you determine the sample size?

You have defined, Ambulatory assessment as a method to study people in their daily life, which consists of a wide range of assessments including momentary self-report, observational assessment, and physiological assessment. One of its’ essential features is observational assessment. How did you meet this feature via online mode? 

We have made revisions throughout the material and methods section. Please see Pages 6-12.

We have added text to state the study design and sample size calculation as: “The present study was a cross-sectional analysis undertaken during September to October 2020. Sample size was calculated according to rules of thumb which was more than 74 (n > 50 + 8m (3 independent variables)). Convenient sampling in the present study recruited 1,001 Thai health professional students. This therefore avoided overfitting problems in regression models”

We used the term “ambulatory assessment” to define a computer-assisted methodology for studying people in their natural environment. Herein, self-reports have been used for gathering data. Students used their own devices to respond to the online questionnaires and tests. We have corrected and added text to clarify this term.

- You have used a number of questionnaires to assess the wellness of the Thai Health Professionals. However, information regarding the scoring directions and Cronbach reliability coefficient scores have not been given. Also, proper direction is required for the Stroop test. 

The validity and reliability of the online Stroop test, IPAQ-SF, PSS-10, MDQ, GAD-7, PHQ-9, MSPSS, and WHOQOL-BREF have been reported in previous studies. We have added references for this information. We have added text to clarify the scoring of questionnaires and tests. Please see Page 9, lines 155-198.

- Page 11, lines 202 to 205: In statistical analysis section, you have mentioned Anxiety, social well-being and quality of life as socio-demographic characteristics, whereas it is different in table 1. (There is confusion in bracket, please, correct it) 

We apologize for the typographical error. We have corrected it as suggested and presented it as: “A regression analysis was performed to determine the impact of COVID-19 awareness (i.e. knowledge, practice, and attitude) and sociodemographic characteristics (i.e. monthly income and accumulated grade point average (GPAX)) on anxiety, social well-being, and quality of life.”

- Page 12, line 217: Write programmes in place of curriculum 

We have changed “curriculum” to “program” as suggested.

- Page 12, line 220: Mention the full form of GPAX while writing first time in the article. Your study also covered year 1 students. Did they get GPAX during the study? You also need to write the full form of other abbreviations such as ‘Walking MET’ (Page 19, line 291). 

We have added “accumulated grade point average” as the full name of “GPAX” in the statistical analysis subsection. We have used “GPAX” according to the questionnaire. GPAX represented in the present study was self-reported by the students. It was GPA for Year 1 students. Thai students know their current GPAX from login the university’s website. We have represented this information by adding footnote in Table 1. We have added the full name of other abbreviations where it was presented the first time in the manuscript as suggested.

- Page 13, lines 230-232: What is the relevant of writing “According to the American College of Sports Medicine (ACSM)’s Guidelines, it indicated that approximately 84% of the students met the conditions for sedentary behaviour” in the result section. 

We have added text to clarify this interpretation in the results section. We also have added reference for the American College of Sports Medicine (ACSM)’s Guidelines. Please see Page 13, lines 243-247.

- Page 15, lines 240 to 247: Description of the questionnaire has already been given in the methodology section. There is no reason to repeat the same in the result section. 

We have removed: “The questionnaires included questions with 3 rating scales that were modified from ... Herein, 1 was minimum and 10 was maximum.” from the results section as suggested.

- Page 16, lines 255 to 256: What does ‘moderate ‘ mean in terms of learning preference? It should be reflected in the methodology section. 

We created the simple numerical rating scale for determining preference on learning under the “new normal.” Preference on learning under the “new normal” was scored from 1 (very unlikely) to 10 (very likely). A score of 5 and 6 of this scale thus suggested that students felt neither “very unlikely” nor “very likely” to learn under the “new normal.” We have added this information in the material and methods section as suggested. We also have changed “moderate” to “at midpoint” for more clarity.

- Page 17, line 277-278: What is high-normal anxiety? Either, it should be severe or normal. 

We have changed “high-normal” to “normal” as suggested.

- Page 19, line 294: It should be no significant relationship 

We have changed this sentence to: “There were no significant relationships between COVID-19 awareness with stress and symptoms of depression.”

- In order to measure the stress level of the students, you have used Perceived Stress. You have written ‘emotional stress’ in some places, which creates confusion. So, write ‘stress’ in place of ‘emotional stress’. We have changed “emotional stress” to “stress”, as suggested.

- Table 5 need more clarity. You have stated, “All three aspects of COVID-19 awareness including knowledge, practice, and attitude were separated in order to reveal their impacts on mood, anxiety symptoms, social well-being, and quality of life among the Thai health professional students when compared to the effects related to the sociodemographic characteristics, i.e. income and GPAX score.” You have 3 aspect of COVID-19 awareness. However, in table 5 only ‘attitude’ has been reported. 

We have added “(Table 5 and S2 Table)” after that sentence and changed the title of Table 5 to “Attitude towards the COVID-19 pandemic and its influence on anxiety, social well-being, and quality of life among Thai health professional students during the early "new normal" informing policy.”

---

## [Editor Report · Decision Letter 1]

20 May 2021

Awareness of COVID-19 influences on the wellness of Thai health professional students: An ambulatory assessment during the early "new normal" informing policy

PONE-D-21-00065R1

Dear Dr. Piya-amornphan,

We’re pleased to inform you that your manuscript has been judged scientifically suitable for publication and will be formally accepted for publication once it meets all outstanding technical requirements.

Kind regards,

Enkeleint A. Mechili

Guest Editor

PLOS ONE

Additional Editor Comments (optional):

All the comments have been addressed accordingly by the authors. The article is ok for publication.
---

## [Editor Report · Acceptance letter]

4 Jun 2021

PONE-D-21-00065R1 

Awareness of COVID-19 influences on the wellness of Thai health professional students: An ambulatory assessment during the early "new normal" informing policy 

Dear Dr. Piya-amornphan:

I'm pleased to inform you that your manuscript has been deemed suitable for publication in PLOS ONE. Congratulations! Your manuscript is now with our production department. 

Kind regards, 

on behalf of

Dr. Enkeleint A. Mechili 

Guest Editor

PLOS ONE